# Educational outcomes of recess in elementary school children: A mixed-methods systematic review

Erin K. Howie[1]*, Kristi L. Perryman[2], Joseph Moretta[2], Laura Cameron[3¤]

1 Department of Health, Human Performance and Recreation, University of Arkansas, Fayetteville, Arkansas, United States of America, 2 Department of Rehabilitation, Human Resources and Communication Disorders, University of Arkansas, Fayetteville, Arkansas, United States of America, 3 University Libraries, University of Arkansas, Fayetteville, AR, United States of America

¤ Current address: Oesterle Library, North Central College, Napierville, Illinois, United States of America
* ekhowie@uark.edu

## Abstract

### Background

Recess provides a key physical activity opportunity for students in school, yet a wide range of recess requirements exist. To design optimal recess policies, the effect of recess on students' educational outcomes must be better understood. Therefore, the purpose of this mixed-method systematic review is to identify and systematically evaluate research on the effects of recess on student educational outcomes, including behavior, cognitive performance and academic achievement.

### Methods

A systematic search of the literature in ERIC (EBSCO), ProQuest Central, PsycINFO, Teacher Reference Center, MEDLINE Complete (EBSCO), and CINAHL Complete was performed through September 2022. Data was extracted from quantitative studies, and reported themes with exemplar quotes were extracted from qualitative studies. The Mixed Method Appraisal Tool (MMAT) was used to assess study quality.

### Results

The search identified 932 articles, of which 13 were included in the review, including 8 quantitative and 5 qualitative studies. Eleven studies were conducted in the United States, and reported sample size of studies ranged from 12 to 11,624. Studies found mixed effects on student behavior, discipline referrals and academic achievement. Qualitative studies reported multiple benefits of recess including increased focus, improved problem solving and academic achievement.

### Conclusions

Overall, evidence suggests positive benefits for behavior and either positive or null benefits of recess on academic achievement. However, evidence is limited by non-controlled study

**Data Availability Statement:** All relevant data are within the paper and its Supporting Information files

**Funding:** The author(s) received no specific funding for this work.

**Competing interests:** The authors have declared that no competing interests exist.

designs and diversity in outcome assessments. Additional quantitative evidence is needed to convince policymakers of the specific evidence supporting recess, but also to advise on the optimal recess policies and practices to improve student learning.

## Introduction

In the United States, only nine states require recess in elementary schools according to the National Association of State Boards of Education: Arizona, Arkansas, Connecticut, Florida, Missouri, New Jersey, Rhode Island, Virginia, and West Virginia [1], with Washington and California most recently deliberating bills. Even within the small group of states with recess legislation, these policies vary widely. For example, Arizona requires two distinct recess periods without specifying duration, Arkansas requires a minimum of 40 minutes of daily recess [1], while the Centers for Disease Control and Prevention (CDC) recommends 20 minutes of daily recess for children [2]. Media reports and advocates cite the benefits of recess for educational outcomes as the driving factor behind these requirements [3]. The American Academy of Pediatrics supports the need for recess with a policy statement describing the importance of recess for social, emotional, physical and cognitive development [4]. However, the body of scientific evidence has not been systematically summarized to help inform current and future policies on optimal durations and implementation of recess practices to achieve maximal educational outcomes. Thus, there is a need to better understand the evidence supporting the impact of recess on academic outcomes.

Recess, according to the CDC is "...a regularly scheduled period in the school day for physical activity and play that is monitored by trained staff or volunteers," and it includes opportunities for physical activity [2]. A large amount of literature has assessed the amount of physical activity obtained during recess e.g. [5,6], including interventions to increase it [7]. There are several benefits of this recess physical activity for children. While there is an association with decreased risks of obesity, diabetes, heart disease, cancer and poor mental health [8], research has found that physical activity specifically during recess may improve children's cardiorespiratory fitness and body composition [9,10]. This highlights that recess can provide a substantial physical activity opportunity for children during the school day.

There has been a growing body of research on the association between physical activity and educational outcomes in children, both immediately after physical activity and after regular exercise [11,12]. Educational outcomes examined have included cognition, on-task behavior, and academic achievement [11–13]. A meta-analysis of 26 studies found physical activity improved classroom behaviors and mathematics and reaching achievement [13], while another meta-analysis of 31 studies found acute physical activity improved attention, but regular physical activity had improvements on attention, executive functions and academic achievement in children [11]. When examining acute effects of physical activity, Hillman et al. found changes in brain activity and cognitive performance in nine to ten year old children following 20 minutes of treadmill walking [14]. Comparatively, to examine the effects of regular physical activity, a nine month afterschool program found improvements in executive functions [15], however a three-year classroom physical activity intervention found no intervention effects on academic achievement [16]. Studies have examined the positive acute effects of physical activity on on-task behavior and attention in the classroom, with a systematic review finding confirming these positive effects [17]. While there are several limitations in previous

research including unknown effects of the duration and intensity of physical activity [11,13,17], evidence suggests physical activity can improve educational outcomes in children.

Besides its opportunity for physical activity, recess may provide a unique physical activity opportunity, that not only includes the physiological response to physical activity, but also benefits of being outdoors, interacting with other children, and enabling creative time during free play (Carlson et al., 2015; CDC, 2020; Hillman et al., 2014; Perryman, et al., 2022). Early research by Piaget classified the developmental stages of play as critical to the intellectual and cognitive development of children [18]. Due to the many benefits of play for development in children, play is guaranteed in the United Nations Convention on the Rights of the Child [19]. As designed, recess includes social interaction, where children have opportunities to develop social skills, practice conflict resolution, and problem-solving skills allowing them to cultivate essential social skills [20]. Activities during recess, such as creative play, can have additional benefits for student outcomes [21]. Recess is typically outdoors, and research on exposure to outdoor nature suggests affective, cognitive, and physical benefits for children [22], leading to a group of Canadian experts creating a position statement on the importance of outdoor active play [23]. Outdoor play, compared to indoor play, includes exposure to nature, sunlight, increased opportunities for risky play, and reduced exposure to potential harms of the internet and screentime which can all influence developmental outcomes [23]. Research suggests that a 15 minute walk outdoors improves cognitive functions such as attention and working memory, while walking indoors, did not [24]. Thus, it is plausible that recess may have effects on educational outcomes in addition to the benefits solely from participating in physical activity.

In order to better understand the effect of recess on educational outcomes, the current literature should first be critically reviewed. Therefore, the purpose of this systematic review was to search and systematically evaluate research on the effects of recess on student educational outcomes, including behavior, cognitive performance and academic achievement. This will help to identify gaps to address in future research and ultimately offer best practice policies for stakeholders and policymakers.

## Methods

Search. The search strategy was registered in PROSPERO [CRD42021221579] and the original protocol is included as Supplementary material 1. The only deviation from the registered protocol was the exclusion of using the GRADE assessment due to the limited search findings and study types. The PRISMA 2020 checklist [25] was used to guide methodology and reporting for this systematic review and the completed checklist is attached as Supplementary Material 2. The search was performed by a health sciences librarian and included the following electronic bibliographic databases: ERIC (EBSCO), ProQuest Central, APA PsycINFO (EBSCO), Teacher Reference Center, MEDLINE Complete (EBSCO), and CINAHL Complete and included "school" and "recess". The search strategy for MEDLINE Complete (EBSCO) was as follows: S1. MH "Schools+", S2. Recess, S3. S1 AND S2 with Limiters: 01-01-2009 to present. The original search was performed in 2019, and thus a 10 year window was used to include recent research. The search was repeated in September 2021 and September 2022 to update search results. The search strategy was adapted for use with other bibliographic databases. No language restrictions were used in the search strategy. Results were limited by date, with results included through September 2022 and published before 2009 excluded. Limiters for source type of academic journal or dissertation were used in ERIC, ProQuest Central, APA PsycINFO, and CINAHL Complete. Following the search, dissertations were removed due to potential duplication with published manuscripts and differing peer-review processes from published peer-reviewed manuscripts. Results were exported to EndNote citation manager,

which was used to identify and remove duplicates. In addition to searching electronic biblio-graphic databases, a hand search was executed to retrieve additional studies for inclusion. The hand search included examining bibliographies of included articles.

The criteria list for study inclusion was based on the following: primary sourced, English language, all elementary (defined as kindergarten through 6th grade) students, recess defined as a regular unstructured break in the school day typically outdoors and including an educa-tional outcome. Educational outcomes considered were defined from previous literature to include student behaviors (i.e. on-task behavior, classroom behavior) cognitive functions (i.e., executive functions, attention, memory, IQ) and academic achievement (i.e., classroom grades, standardized tests, classroom behavior), and could include perceived changes in these out-comes from qualitative studies. All study designs were included. Studies examining a particular population sub-set (e.g., students with autism) were excluded. Additionally, studies of inter-ventions where recess was manipulated and no longer unstructured activity (e.g., a fitness pro-gram during recess or an educational program) or part of a multicomponent study where the individual effects of recess were not separated were excluded unless the effects of recess alone were reported.

Two reviewers independently assessed titles, abstracts and full-articles for inclusion. Dis-agreements were settled by a consensus or when necessary a third senior reviewer. Relevant PICO (population, intervention, comparison and outcomes) information from quantitative studies was extracted by two reviewers separately, and then reviewed until consensus was reached. Reported themes with exemplar quotes were extracted from qualitative studies [26]. A senior researcher (Author EKH) supervised the review and facilitated discussion of disagree-ments. Study quality was assessed by two reviewers in consultation. Due to the potential for numerical rating systems to under identify bias, subjective interpretation is recommended [27]. Due to the heterogeneity in study designs, the Mixed Method Appraisal Tool (MMAT) was used for qualitative and quantitative studies. This rating system has been widely used across disciplines and guides reviewers to assess internal validity of multiple study types [28]. Within person studies were evaluated as quantitative non-randomized studies. The tool devel-opers discourage the use of overall scoring but advise for detailed presentation of the ratings of each criterion, thus individual scores for each item are reported and overall bias of studies is discussed.

## Results

### Summary of search process

The PRISMA flow diagram can be seen in Fig 1. After removing duplicates, 658 articles were found and 50 were added through a hand search of reference lists. 671 articles were excluded after examining titles, including 71 dissertations or theses by reported publication type. Thirty-seven full articles were reviewed and 24 were excluded, resulting in 13 included articles. The primary reasons for exclusion were not examining an association between recess and edu-cational outcomes, recess was not separately examined from other physical activity opportuni-ties, or the study included an additional recess intervention.

**Study details.** There was wide heterogeneity in studies, and a summary of quantitative studies can be seen in Table 1 and qualitative studies in Table 2. Study publication dates ranged from 2009 to 2021. Study designs included qualitative (n = 5), quantitative descriptive (n = 3), and quantitative non-randomized (n = 5). All but two of the studies were conducted in the United States (Texas n = 1, New York n = 1, Kentucky = 2, Mississippi n = 1, multiple states n = 5, not reported n = 1); the international studies were from Turkey and Greece. Quantita-tive studies included students from kindergarten through 6th grade. Sample size of quantitative

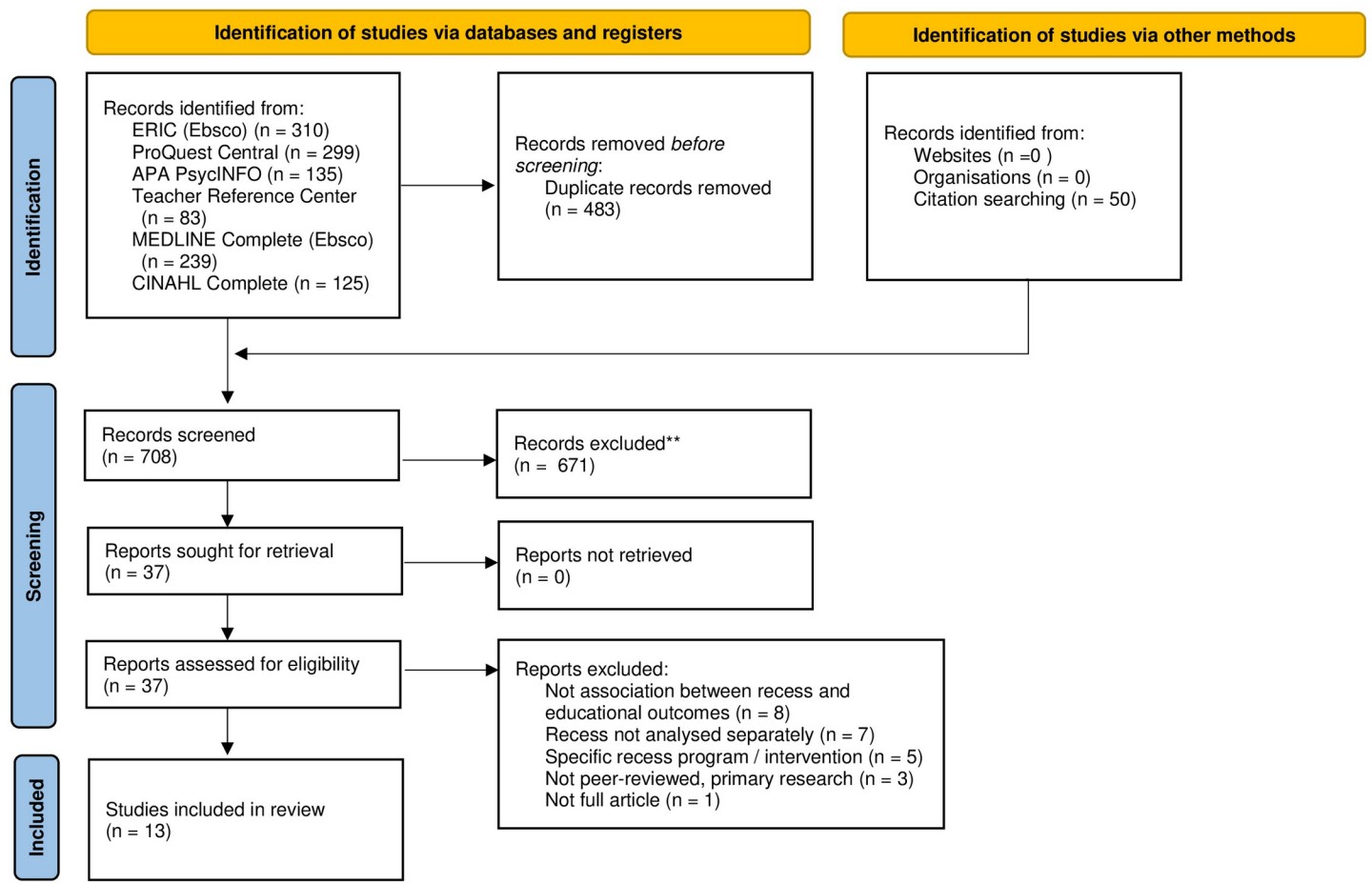

**Fig 1. PRISMA flow chart of study selection.** The number of studies identified, screened and included through the systematic review process.

studies ranged from 12 to 11,624 participants with one study not reporting included sample size [29]. Three studies were secondary analyses of the Early Childhood Longitudinal Studies–Kindergarten class of 1998–1999 (ECLS-K). In the experimental studies, the majority were pre-post designs. Two tested the effects of increasing the amount and frequency of recess to two, 15 minute recesses [30,31]. Quantitative studies on the LiiNK project, a specific recess intervention where four,15 minute recess periods were implemented, were excluded due to the LiiNK program also including a character development intervention [32,33]. Another two studies evaluated the acute effects after a single recess period [34,35]. One study examined the timing of discipline referrals related to recess scheduling [31]. Qualitative study participants included teachers, principals, parents and students. Three examined perceptions of recess benefits in general [36–38], one examined perceptions of the LiiNK program which increased recess to four, 15 minute recesses per day [39], and one compared perceptions of regular recess to a structured activity program Let Grow Play Club [40].

**Relationships between recess and educational outcomes.** Of the quantitative studies, 5 studies examined some type of behavior which included observed on-task behavior,[33,35] teacher rated classroom behavior [42,44], and discipline referrals[30,31]. In the ECLS-K data, Barros et al found that having some recess was associated with better teacher reported behavior compared to having minimal or no daily recess [42]. Stapp examined on-task behavior

**Table 1. Summary of quantitative studies.**

| Study first author, date | Study type | Country; State | n | Grades | Recess Duration & frequency | Outcome Measure | Findings |
|---|---|---|---|---|---|---|---|
| Dills, 2011 [29] | Longitudinal (ECLS-K) | US; multiple | Not reported | K through 5th | Average 133.4 min per week in K to 89.4 min in 5th | Reading and math scores from ECLS-K reading assessments | No effect of recess on reading or math scores. |
| Brez, 2017 [34] | Experimental (Acute; pre-post, no control) | US; NY | 99 | 3rd-5th | Not specified | Sustained attention (letter canceling task), Creativity (Alternate Uses Task) | Sustained attention improved following recess period. |
| Stapp, 2018 [35] | Experimental (acute; pre-post, not control) | US; [41] MS | 12 | 5th | 25 minutes | On-task behavior (observation) | On-task behavior increased following recess |
| Erwin, 2019 [30] | Experimental (chronic; pre-post, no control) | US; KY | 728 | K-6th | 2 x 15 minutes (increased from 1 x 15 minutes) | Discipline Referrals, Academic achievement (MAP test) | Discipline referrals increased and math achievement improved with increased recess. |
| Fedewa, 2021 [31] | Experimental (acute; pre-post, no control) | US; KY | 607 | K-6th | 2 x 15 minutes (increased from 1 x 15 minutes) | Discipline Referrals | Discipline referrals increased with more time elapsed since recess period |
| Barros, 2009 [42] | Cross-sectional sample within Longitudinal (ECLS-K) | US; multiple | 10,301–11,624 | 3rd (including some 2nd and 4th graders) | 70% had 15 min or more of recess per day | Teacher rated group classroom behavior | Students with some recess were in classrooms with better teacher reported behavior compared to those with no or minimal recess. |
| Yesil Dagli, 2012 [43] | Cross-sectional sample within Longitudinal (ECLS-K) | US; multiple | 3,951 | K | 79% had daily recess; 67% had 16–30 minutes per day | Reading scores from ECLS-K reading assessments | No relationship between recess frequency and recess duration separately with reading scores. Different combinations of recess frequency and duration resulted in higher reading scores. |
| Massey, 2021 [44] | Cross-sectional | US; multiple | 352 | 3rd & 5th | Mean 29.5 minutes/recess | Classroom behavior (BASC-3) | Recess quality was associated with adaptive classroom behavior, executive functioning problems, resilience, and emotional self-control. Recess time was associated with lower levels of externalizing problems and bullying. |

immediately after a 25 minute recess, compared to before recess and found that on-task behavior improved but did not have a control comparison [35]. Using a paired-test for the 12 included participants, time on-task increased from 36.6% to 70.3%. Massey et al found better recess quality was associated with some aspects of behavior [44]. Two studies utilizing the same natural experiment found that discipline referrals increased when recess doubled from one, 15 minute recess to 2, 15 minute recesses utilizing mixed-effects models [30], but more discipline referrals were made with increased time elapsed from the scheduled recess [31].

Three studies examined the effect of recess on academic achievement. Yesil Dagli found no relationship between recess duration or frequency with reading scores in the ECLS-K kindergarten sample [43], Dills found no effects of recess time on reading or math achievement longitudinally in the ECLS-K cohort [29], while Erwin et al. found improved math achievement scores but not reading after recess increased from one recess to two 15 minute recesses [30]. Other outcomes included cognitive tasks of sustained attention and creativity [34]. Sustained attention, but not creativity, improved following a single recess period among 3rd and 5th graders [34].

Of the themes discussed in the qualitative studies relating to the perceived benefits of recess, participants reported multiple perceived benefits. Three studies described focus [36,39,40], and three described benefits in problem solving skills [36,39,40]. Two studies described better

**Table 2. Summary of qualitative studies (n = 5).**

| | Country/ State | Participants | Recess duration | Perceived Educational Outcomes | Representative Quotes |
|---|---|---|---|---|---|
| Martin, 2018 [36] | US; KY, TN, TX | 16 college students, teachers, parents | N/A | Classroom behavior, focus, problem-solving | "Time away from the classroom can foster creative thinking when returning to problems to solve in a classroom. Also, sunshine and the great outdoors can lift anyone's spirit."–parent (only parent quotes provided) |
| Bauml, 2020 [39] | US; Texas | 17 teachers; K, 1st, 2nd, PE | 4 x 15 min; LiiNK project | Sustaining Focus; Academics; Creativity; Problem Solving | "I think it's maintained things. It hasn't been detrimental. And, see, and some people thought, because we were taking the time away, that [grades] would go down, but no, I don't see that at all."–K teacher; "They have to be creative,"– 2nd teacher |
| Ozkal, 2020 [37] | Turkey | 1 teachers and administrators; primary and secondary school | Not specified; Legislated minimum of 15 minutes | Cognitive; Behavioral; Academic learning; Negative effects | "I mean recess is a period in which learning actually takes place. Training and education are not provided only during class hours, learning also takes place during recess. Some of the students can even discuss what they have learned in class during the break. . ..–primary teacher; "Students can be distracted by recess, they can be focused in class only for 3–5 minutes; if they really enjoy the game outside, they are really distracted and would prefer to be outside".–primary teacher; They come back from recess happier. They come back with smiling faces. . .–primary teacher |
| Parrott, 2020 [40] | US; New York | 47 students, 6 teachers; observations of recess | 40 minute recess (as control to 60 minute Let Grow Play Club) | Focus; Problem solving | "I think my kids perform better in the afternoon, after the longer recess, than in the morning. . .[a] sustained period of that activity, I think, provides them a longer period of attention when they come back into the room."–K special education teacher; ". . .I think it offers many opportunities for them to solve problems, without me telling them what they should be doing, when they have the time to figure it out themselves.–NS teacher |
| Prompona, 2020 [45] | Greece | 82 students 1st-6th | 4 recess periods for a total of 55 minutes (25, 15, 10, 10) | Creation-imagination | It's nice to make up stories. To plan things, to have new ideas! It's like scripting new films, like being movie directors!– 3rd grader |

academic achievement as a perceived benefit [37,39]. Two mentioned creativity [38,39], and two mentioned improvements to behavior [36,37]. Only one study conducted by Ozkal in Turkey reported a negative theme, that students become distracted by preferring to be out at recess [37].

**Study quality.** Results from the MMAT assessment can be found in Table 3. All but one qualitative study was of good methodological quality for each criterion. The one mixed-methods study rated highly for the qualitative component but did not integrate the quantitative and qualitative components towards an overall research purpose. The quantitative non-randomized studies had varying quality mostly due to lack of reporting of sample characteristics and missingness in outcome data. Two studies were limited in their ability to answer the proposed research questions, one due to a small sample size and one due to the combination of lunch and recess in the exposure. The quantitative descriptive studies were at minimal risk of bias.

## Discussion

This review examined the relationships between recess and educational outcomes in elementary students. Overall, quantitative evidence suggests positive benefits for behavior and either positive or null benefits of recess on academic achievement. Qualitative reports from teachers describe multiple benefits including problem solving and focus. However, evidence is limited by non-controlled study designs and diversity in outcome assessments.

**Table 3. Mixed Method Appraisal Tool (MMAT) assessment of study quality.**

| First Author, Pub Date | All Studies | | Qualitative Studies | | | | | Quantitative Non-randomized | | | | | Quantitative Descriptive | | | | | Mixed-Methods | | | | |
|---|---|---|---|---|---|---|---|---|---|---|---|---|---|---|---|---|---|---|---|---|---|---|
| | S1 | S2 | 1.1 | 1.2 | 1.3 | 1.4 | 1.5 | 3.1 | 3.2 | 3.3 | 3.4 | 3.5 | 4.1 | 4.2 | 4.3 | 4.4 | 4.5 | 5.1 | 5.2 | 5.3 | 5.4 | 5.5 |
| Martin, 2018 [36] | Y | Y | Y | N | N | N | N | | | | | | | | | | | | | | | |
| Bauml, 2020 [39] | Y | Y | Y | Y | Y | Y | Y | | | | | | | | | | | | | | | |
| Ozkal, 2020 [37] | Y | Y | Y | Y | Y | Y | Y | | | | | | | | | | | | | | | |
| Parrott, 2020 [40] | Y | Y | Y | Y | Y | Y | Y | | | | | | Y | Y | Y | Y | Y | N | N | N | N | Y |
| Prompona, 2020 [45] | Y | Y | Y | Y | Y | Y | Y | | | | | | | | | | | | | | | |
| Dills, 2011 [29] | Y | Y | | | | | | Y | Y | N | Y | Y | | | | | | | | | | |
| Brez, 2017 [34] | Y | N | | | | | | C | C | Y | Y | Y | | | | | | | | | | |
| Stapp, 2018 [35] | Y | N | | | | | | N | Y | Y | NA | Y | | | | | | | | | | |
| Erwin, 2019 [30] | Y | Y | | | | | | Y | Y | N | Y | Y | | | | | | | | | | |
| Fedewa, 2021 [31] | Y | Y | | | | | | Y | Y | Y | Y | Y | | | | | | | | | | |
| Barros, 2009 [42] | Y | Y | | | | | | | | | | | Y | Y | Y | C | Y | | | | | |
| Yesil Dagli, 2012 [43] | Y | Y | | | | | | | | | | | Y | Y | Y | Y | Y | | | | | |
| Massey, 2021 [44] | Y | Y | | | | | | | | | | | Y | Y | Y | Y | Y | | | | | |

Y Yes, N No, C Can't determine, NA not applicable, **S1**: Are there clear research questions? **S2**: Do the collected data allow to address the research questions? **1.1**: Is the qualitative approach appropriate to answer the research question? **1.2**: Are the qualitative data collection methods adequate to address the research question? **1.3**: Are the findings adequately derived from the data? **1.4**: Is the interpretation of results sufficiently substantiated by data? **1.5**: Is there coherence between qualitative data sources, collection, analysis and interpretation? **3.1**. Are the participants representative of the target population? **3.2**. Are measurements appropriate regarding both the outcome and intervention (or exposure)? **3.3**. Are there complete outcome data? **3.4**. Are the confounders accounted for in the design and analysis? **3.5**. During the study period, is the intervention administered (or exposure occurred) as intended? **4.1**. Is the sampling strategy relevant to address the research question? **4.2**. Is the sample representative of the target population? **4.3**. Are the measurements appropriate? **4.4**. Is the risk of nonresponse bias low? **4.5**. Is the statistical analysis appropriate to answer the research question? **5.1**. Is there an adequate rationale for using a mixed methods design to address the research question? **5.2**. Are the different components of the study effectively integrated to answer the research question? **5.3**. Are the outputs of the integration of qualitative and quantitative components adequately interpreted? **5.4**. Are divergences and inconsistencies between quantitative and qualitative results adequately addressed? **5.5**. Do the different components of the study adhere to the quality criteria of each tradition of the methods involved?

Studies found that either having more recess [35,42] or better quality recess [44] was associated with better student behavior, however, the magnitude of the effect cannot be interpreted from the few studies, diverse study designs and measures, and consistent reporting of statistical findings. This may be due to several underlying cognitive or executive function mechanisms such as neuroelectric changes in response to acute bouts of exercise [46] or changes in brain health, structure and function [47]. However, another study found that the number of discipline referrals increased with a doubling of recess time [30]. This is likely due to a high percentage of discipline referrals occurring during recess; thus, increasing the duration of recess would increase the amount of discipline referrals. Additionally, there has been variety in how student behavior has been assessed from official discipline referrals to teacher reported behavior to observed classroom behavior. The study which evaluated the implementation of two, 15 minute recesses in Kentucky, also found that the occurrence of discipline referrals increased as time since the last recess elapsed [31]. This has important implications for principals scheduling recess. It may be best to reduce discipline referrals by spreading recess throughout the day to minimize long durations of school time without recess. Additionally, for schools, districts, and states considering increasing the amount of recess, it may be prudent to include positive behavior or conflict resolution curriculums to help mitigate increases in discipline referrals. Playworks is a non-profit organization that provides training, staffing and resources to improve the quality of recess that has shown to improve physical activity in girls [48], teacher

perceptions of safety and inclusion, and reduced teacher perceptions of bullying and time to transition to learning activities [49]. Other strategies such as schoolyard greening [50], or token economy incentives [51], might improve play and reduce negative behaviors and ultimately improve educational outcomes for students.

The quantitative studies did not specifically examine a dose response between recess duration and outcomes, however, the experimental studies that had improved outcomes examined a 25 minute recess period [35], or two 15 minute recess periods [30]. Additionally, in a study where greater recess time was associated with lower externalizing problems, the average recess time was 30 minutes with a range from 20 to 60 minutes [44]. Additionally, three of the qualitative studies where recess duration was specified were all 40 minutes of daily recess or more, with two breaking up the total time into shorter periods [39,45]. While the evidence is limited, this suggests that there may be additional educational benefits of recess length longer than the CDC recommended 20 minutes per day, but that this can be broken up into shorter recess periods.

Only three studies directly examined the effect of recess on academic achievement, with studies utilizing the ECLS-K cohort finding no relationship between recess and academic achievement measured as math and reading achievement on the National Assessment of Educational Progress [29,43], and one study finding an increase in math achievement on the Measure of Academic Progress (MAP) standardized test with two, 15 minute recesses per day [30]. This is important as often increasing recess time is considered to take away from classroom learning time; however, the additional time allocated to recess was not shown to reduce academic achievement. Other academic related outcomes such as sustained attention and creativity [34] have been shown to improve immediately following a recess period. More studies examining the effects of recess on these cognitive outcomes, both acutely and longer term, may help to bridge the gap in understanding the impact of recess on more distal academic achievement [12]. Physical activity has been shown to improve cognitive performance, particularly executive functions, even after 20 minutes of walking in children [14]. In addition to the physical benefits of physical exercise, recess, which also includes social interactions, games, and opportunities for creative play, may have additional cognitive benefits compared to non-cognitively engaging physical activity [21]. Additional research examining the acute effects of recess, and some of the contextual factors of recess, on both cognitive performance and behavior may provide intermediary mechanisms to influence ultimate academic achievement. Potential contextual factors that may influence recess could be the intensity of physical activity, type of play, peer interactions, and teacher involvement [21,52].

Other similar reviews conducted have differed in their methodology. One recent systematic review searched for in-school play opportunities, without focusing on recess [53]. However, they were unable to find studies of other in-school play opportunities and qualitatively reviewed 20 studies on recess. They concluded that recess was beneficial for student behavior with mixed outcomes for academic achievement; however, they did not include a discussion of differences in outcome measures and the review included recess interventions where play was structured such as the Playworks program. Another review specifically looking at recess included nine studies [54]. While they excluded recess interventions, they did not conduct a study quality assessment and included studies on the physical benefits of recess physical activity. They concluded that recess does not have a negative impact on academic achievement and has positive benefits on student behavior. The strengths of the current systematic review were pre-registration in PROSPERO, systematic approach to reviewing quantitative and qualitative literature, in addition to a systematic review of study quality.

The studies included in this review were quasi-experimental and most lacked a control condition. This is expected as changes to recess durations often involve large scale policies or school district changes that do not lend themselves to randomized control trials. Leading

experts on school physical activity interventions have advocated for the need to include context in both the design and evaluation of programs and policies [55]. Importantly, they emphasize the need to consider other rigorous study designs to randomized control trials. Natural experiments may be helpful to examine the impact of changes in recess policies [56]. Additionally, as the effects of recess may be acute, within subject designs may help to elucidate some of these acute cognitive and behavioral impacts immediately following recess. While the qualitative studies reported generally favorable perceptions of recess by teachers and stakeholders, additional quantitative evidence is needed to convince policymakers of the specific evidence supporting recess, but also to advise on the optimal recess policies and practices to improve student learning.

This review only examined English language, peer-reviewed, primary research articles. Many commentaries and dissertations from the education field were not included, though not all met inclusion and exclusion criteria, to avoid duplication with peer-reviewed articles and maintain a consistent standard of peer-reviewed evidence. To avoid publishing bias, it may be beneficial for students and their mentors to produce high quality, publishable research to submit before or after graduation. Additionally, this review did not include widely heterogenous intervention studies where recess was manipulated in order to examine the effects of standard recess. Some of these interventions [48,51] have shown to have positive effects on student behaviors, and many are included within wider school-based physical activity interventions [57]. As recess has a large potential reach, many schools may not have the resources to implement staff or equipment intensive recess programs. However, they may be able to schedule additional recess time, if they are provided evidence on the optimal scheduling and tangible, meaningful outcomes for students' academic achievement. Furthermore, these interventions may provide information on some of the contextual factors, such as teacher involvement, peer behaviors, and loose equipment needs that can help to guide practitioners when resources are available.

## Conclusion

This systematic review found limited evidence that recess may be associated with improved student behavior, with no negative effects on academic achievement. The optimal daily recess duration may be greater than 20 minutes, with multiple recess periods in a day. However, the current evidence is heterogenous and limited by methodological rigor and outcome assessments. Researchers should conduct natural experiments or other controlled study designs to further clarify the effects of recess quantity and quality on student educational outcomes, including acute responses in cognitive function. Though additional evidence on the effects of the optimal recess dosage is still needed to maximize the potential improvements to not only student health, but also educational outcomes, educational stakeholders such as superintendents, principals, or legislators should consider implementing CDC recommended 20 minutes of daily recess and potentially more.

## Supporting information

**S1 Checklist. PRISMA 2020 checklist.**
(DOCX)

**S1 File.**
(PDF)

## Author Contributions

**Conceptualization:** Erin K. Howie, Kristi L. Perryman.

**Data curation:** Erin K. Howie, Joseph Moretta, Laura Cameron.

**Formal analysis:** Erin K. Howie, Joseph Moretta, Laura Cameron.

**Investigation:** Erin K. Howie, Laura Cameron.

**Methodology:** Erin K. Howie, Laura Cameron.

**Project administration:** Erin K. Howie.

**Supervision:** Erin K. Howie, Kristi L. Perryman.

**Writing – original draft:** Erin K. Howie.

**Writing – review & editing:** Kristi L. Perryman, Joseph Moretta, Laura Cameron.

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
