## [Decision Letter · Decision Letter 0]

14 Aug 2023

PONE-D-23-20058Educational outcomes of recess in elementary school children: A mixed-methods systematic reviewPLOS ONE

Dear Dr. Howie,

Thank you for submitting your manuscript to PLOS ONE. After careful consideration, we feel that it has merit but does not fully meet PLOS ONE’s publication criteria as it currently stands. Therefore, we invite you to submit a revised version of the manuscript that addresses the points raised during the review process.

ACADEMIC EDITOR: Please insert comments here and delete this placeholder text when finished. Be sure to:Respond to the reviewers comments within 2 weeks==============================

We look forward to receiving your revised manuscript.

Kind regards,

Felix Apiribu, Ph. D., MPhil

Academic Editor

PLOS ONE

Reviewers' comments:

Reviewer's Responses to Questions

**Comments to the Author**

1. Is the manuscript technically sound, and do the data support the conclusions?

Reviewer #1: Yes

Reviewer #2: Partly

2. Has the statistical analysis been performed appropriately and rigorously? 

Reviewer #1: N/A

Reviewer #2: I Don't Know

3. Have the authors made all data underlying the findings in their manuscript fully available?

Reviewer #1: Yes

Reviewer #2: Yes

4. Is the manuscript presented in an intelligible fashion and written in standard English?

Reviewer #1: Yes

Reviewer #2: Yes

5. Review Comments to the Author

Reviewer #1: An interesting and important study. The paper would have benefited from inclusion of some statistical data to exemplify the degree to which interventions affects outcomes because this was largely absent. A few additional suggestions to improve the paper.

Page 6 line 95, it may be useful to elaborate a little on the concept of playing outdoors compared to indoors as this may be pivotal to your argument.

Page 11 Line 168. Yu mention that your studies included through to grade 6, but your methods say grade 5 (page7 line 123), this needs to be reconciled.

Page 19 and 20, discuss aspects that suggest shorter recess periods (discipline referrals) and longer recess periods ((educational benefits). It would be nice to see these two concepts integrated into a single argument to provide a more structured conclusion and opinion.

Reviewer #2: I am reviewing this manuscript from the lens of my expertise, evidence synthesis information retrieval. I am concerns about the lack of detail about the search which makes it irreproducible.

PRISMA Flow should have total # of results form electronic databases pre and post deduplication. There isn't enough detail about the search strategy to be reproducible.

A search in ERIC (Ebsco) for

( DE "Recess Breaks" OR TI (Recess) OR AB (Recess) ) AND ( TI (School*) OR AB (School*) )

limited to 2009 to present gave me 413 results on Aug. 2, 2023. That's just one database which has me confused about the total number of unique search results.

See PRISMA guidelines for each database total needing to be represented in the flow diagram.

Limiting to English-only should be listed as a limitation. The 2009 publication date limit should be justified and/or listed as a limitation.

It's unclear how duplicates were removed pre-screening. Was it a citation manager like Zotero? Covidence? Please indicate. I am happy to see that handsearching was conducted to find things not showing up in the electronic search strategy.

In the body of the manuscript, the search strategy is described but without enough information to make it reproducible.

MH is not a field code in ERIC so the depiction of the search doesn't make sense. MH is likely reflecting field codes in either CINAHL or Medline.

Electronic search strategies should consist of relevant subject headings and search terms (e.g., titles and abstract fields). In ERIC, the subject heading for recess is DE Recess Breaks. Authors should disclose which platform was used to search PsycINFO (Ovid? Ebsco?).

6. PLOS authors have the option to publish the peer review history of their article (what does this mean?). If published, this will include your full peer review and any attached files.

Reviewer #1: No

Reviewer #2: No

---

## [Author Response · Author response to Decision Letter 0]

28 Aug 2023

Overall, we thank both Reviewers’ for their careful review and thoughtful critiques. We believe the detailed changes as detailed below, including a more complete description of the search strategy, strengthen the review. 

Reviewer #1: An interesting and important study. The paper would have benefited from inclusion of some statistical data to exemplify the degree to which interventions affects outcomes because this was largely absent. A few additional suggestions to improve the paper.

Response to Review: We thank the Reviewer for their interest and positive view of our study. With the few quantitative studies of various designs, we were not able to conduct a complete meta-analysis of the intervention effects as there were only four experimental studies which had wide variation in the completeness of reporting for statistical interpretation. We have added description of individual statistical findings when it would aid in interpreting, e.g. “Using a paired-test for the 12 included participants, time on-task increased from 36.6% to 70.3%.” We have added this limitation to the Discussion, “however, the magnitude of the effect cannot be interpreted from the few studies, diverse study designs and measures, and consistent reporting of statistical findings.

Page 6 line 95, it may be useful to elaborate a little on the concept of playing outdoors compared to indoors as this may be pivotal to your argument.

Response to Review: We have added to the discussion of the benefits of outdoor exposure, in addition to the previous empirical evidence directly comparing indoor and outdoor walking.” Outdoor play, compared to indoor play, includes exposure to nature, sunlight, increased opportunities for risky play, and reduced exposure to potential harms of the internet and screentime which can all influence developmental outcomes [54].”

Page 11 Line 168. Yu mention that your studies included through to grade 6, but your methods say grade 5 (page7 line 123), this needs to be reconciled.

Response to Review: We apologize for the mistake. Elementary was defined as kindergarten through 6th grade and this has been clarified in the Methods. 

Page 19 and 20, discuss aspects that suggest shorter recess periods (discipline referrals) and longer recess periods ((educational benefits). It would be nice to see these two concepts integrated into a single argument to provide a more structured conclusion and opinion.

Response to Review: Only one study (with 2 reports) included in the review found an increase in discipline referrals, with 3 studies suggesting positive effects on student behavior, and no studies have directly measured both behavior and discipline referrals. We have included a practical discussion of these findings, “This has important implications for principals scheduling recess. It may be best to reduce discipline referrals by spreading recess throughout the day to minimize long durations of school time without recess. Additionally, for schools, districts, and states considering increasing the amount of recess, it may be prudent to include positive behavior or conflict resolution curriculums to help mitigate increases in discipline referrals.”

Reviewer #2: I am reviewing this manuscript from the lens of my expertise, evidence synthesis information retrieval. I am concerns about the lack of detail about the search which makes it irreproducible.

Response to Review: The authors appreciate the Reviewer’s expertise and feedback, and have incorporated revisions to the manuscript to address the reviewers concerns. Notably, the PRISMA Flow diagram has been updated with more complete information, including the number of unique results located within each database pre-duplication.

The methods section has been expanded to make the search strategies used more explicit and, in turn, more reproducible. We thank the reviewer for catching the error in our provided search statement, which was the search for MEDLINE Complete, not ERIC. Other errors, such as the missing platform name used to search APA PsycINFO, have also been fixed.

PRISMA Flow should have total # of results form electronic databases pre and post deduplication. There isn't enough detail about the search strategy to be reproducible.

Response to Review: We have updated the PRISMA flow chart to include the total results from each database.

A search in ERIC (Ebsco) for

( DE "Recess Breaks" OR TI (Recess) OR AB (Recess) ) AND ( TI (School*) OR AB (School*) )

limited to 2009 to present gave me 413 results on Aug. 2, 2023. That's just one database which has me confused about the total number of unique search results.

Response to Review: We have more clearly defined the search strategy in the Methods section, along with the search results by database. 

See PRISMA guidelines for each database total needing to be represented in the flow diagram.

Response to Review: We have updated our PRISMA flow chart to include the results from each database.

Limiting to English-only should be listed as a limitation. The 2009 publication date limit should be justified and/or listed as a limitation.

Response to Review: We have clarified in the methods that the 2009 date was included from the first search in 2019, which was later repeated to update the results. We have added the English-only limitation to the Discussion. 

It's unclear how duplicates were removed pre-screening. Was it a citation manager like Zotero? Covidence? Please indicate. I am happy to see that handsearching was conducted to find things not showing up in the electronic search strategy.

Response to Review: Duplicates were removed using a citation manager, which we have added ot the Methods section. 

In the body of the manuscript, the search strategy is described but without enough information to make it reproducible.

Response to Review: We have added detail to the search strategy in the Methods section. 

MH is not a field code in ERIC so the depiction of the search doesn't make sense. MH is likely reflecting field codes in either CINAHL or Medline.

Response to Review: Our apologies for the mis-identified example search (MEDLINE Complete (EBSCO). We have more specifically identified the databases searched. 

Electronic search strategies should consist of relevant subject headings and search terms (e.g., titles and abstract fields). In ERIC, the subject heading for recess is DE Recess Breaks. Authors should disclose which platform was used to search PsycINFO (Ovid? Ebsco?).

Response to Review: We have specified the APA PsycINFO (EBSCO) platform was used.

---

## [Decision Letter · Decision Letter 1]

31 Oct 2023

Educational outcomes of recess in elementary school children: A mixed-methods systematic review

PONE-D-23-20058R1

Dear Dr. Howie,

We’re pleased to inform you that your manuscript has been judged scientifically suitable for publication and will be formally accepted for publication once it meets all outstanding technical requirements.

Kind regards,

Francisco Wilker Mustafa Gomes Muniz

Academic Editor

PLOS ONE

Additional Editor Comments (optional):

Reviewers' comments:

Reviewer's Responses to Questions

**Comments to the Author**

1. If the authors have adequately addressed your comments raised in a previous round of review and you feel that this manuscript is now acceptable for publication, you may indicate that here to bypass the “Comments to the Author” section, enter your conflict of interest statement in the “Confidential to Editor” section, and submit your "Accept" recommendation.

Reviewer #1: All comments have been addressed

Reviewer #2: All comments have been addressed

2. Is the manuscript technically sound, and do the data support the conclusions?

Reviewer #1: Yes

Reviewer #2: Yes

3. Has the statistical analysis been performed appropriately and rigorously? 

Reviewer #1: N/A

Reviewer #2: I Don't Know

4. Have the authors made all data underlying the findings in their manuscript fully available?

Reviewer #1: Yes

Reviewer #2: Yes

5. Is the manuscript presented in an intelligible fashion and written in standard English?

Reviewer #1: Yes

Reviewer #2: Yes

6. Review Comments to the Author

Reviewer #1: I accept the response regarding absence of statistical information and thank you for addressing the remaining queries.

Reviewer #2: Thank you for the opportunity to review the revision of this manuscript. I am pleased to see that the team modified their PRISMA flow diagram to reflect key details (result totals, platforms). Additionally the methods section was expanded to include more detail about the search strategies. The syntax is now correctly labeled for ERIC. Thank you for clarifying deduplication details. In the future, please search with multiple metadata fields and OR them together. For example in Medline (Ebsco), it'd be (MH Schools+ OR TI school* OR AB school*) AND (TI recess OR AB recess). Since you did citation searching by-hand, I am hopeful that any items missed during your electronic search strategy were caught via your complementary handsearching. Also, it's a PRISMA 2020 rule to include the full electronic search strategy (not handsearching) for all databases. I recommend including those in the appendix or supplemental files.

7. PLOS authors have the option to publish the peer review history of their article (what does this mean?). If published, this will include your full peer review and any attached files.

Reviewer #1: No

Reviewer #2: No

---

## [Editor Report · Acceptance letter]

13 Nov 2023

PONE-D-23-20058R1 

Educational outcomes of recess in elementary school children: A mixed-methods systematic review 

Dear Dr. Howie:

I'm pleased to inform you that your manuscript has been deemed suitable for publication in PLOS ONE. Congratulations! Your manuscript is now with our production department. 

Kind regards, 

on behalf of

Dr. Francisco Wilker Mustafa Gomes Muniz 

Academic Editor

PLOS ONE